# Cognition, physical function, and life purpose in the rural elderly population: A systematic review protocol

**Hércules Lázaro Morais Campos**[1]*, **Elisa Brosina De Leon**[2], **Ingred Merllin Batista de Souza**[3], **Anna Quialheiro**[4], **Elizabete Regina Araújo de Oliveira**[5]

1 Professor at the Physiotherapy Department of the Federal University of Amazonas/ Institute of Health and Biotechnology (UFAM/ISB), Amazonas, Brazil, 2 Stricto Sensu Post-graduation Program in Human Movement Sciences, Faculdade de Educação Física e Fisioterapia, Professor at the Universidade Federal do Amazonas, Manaus, Amazonas, Brazil, 3 Department of Physical Therapy, Speech-Language Pathology, and Audiology and Occupational Therapy, Faculty of Medicine at University of São Paulo, São Paulo, Brazil, 4 Health and Human Movement Unit, Instituto Politécnico de Saúde do Norte—CESPU, Vila Nova de Famalicão, Portugal, 5 Professor of the Post-graduate Program in Collective Health of the Federal University of Espírito Santo, Vitória, Brazil

* herculeslmc@hotmail.com

**Data Availability Statement:** No datasets were generated or analysed during the current study. All

## Abstract

### Introduction

Aging in rural settings worldwide, from the perspective of cognition, physical function, and life purpose essential constructs for a prosperous old age, still needs comprehensive discussion. This systematic review protocol aims to highlight the prevalence of cognitive decline, physical functioning, and life purpose in older adults aging in rural community settings.

### Methods and analysis

We will include cross-sectional studies published until April 2023 found in 8 databases: Embase, MEDLINE, LILACS, PsycINFO, Scopus, SciELO, and Web of Science. Ryyan software will be used for the first selection, and the Observational Study Quality Evaluation (OSQE) will assess methodological quality and risk of bias. Primary analysis will involve titles and abstracts using MeSH descriptors such as "Physical functioning," "Cognition," "Cognitive function," "Life purpose," "Elderly," "Older," "Rural aging," "Rural population," "Communities, rural," "Distribution, rural spatial," "Medium communities," "Rural settlement," "Small community." If necessary, secondary analysis will include a complete reading of selected articles by two blinded reviewers, confirmed by a third person. Publication bias will be assessed using cross-sectional analytical study quality. Sensitivity analyses will identify manuscripts significantly influencing combined prevalence of endpoints.

relevant data from this study will be made available upon study completion.

**Funding:** Fundação de Amparo à Pesquisa do Estado do Amazonas, EDITAL N. 012/2021 – POSGFE Ms Hércules Lázaro Morais Campos anf Research scholarship from the Amazonas State Research Support Foundation - Brazil (FAPEAM). The funders had no role in study design, data collection and analysis, decision to publish, or preparation of the manuscript.

**Competing interests:** The authors have declared that no competing interests exist.

## Introduction

Aging within the rural context takes place in various ways globally. The literature increasingly emphasizes that individuals aging in this unique setting exhibit different epidemiological and health indicators than seniors in urban contexts [1].

Controversies arise regarding elderly individuals aging in rural settings, with challenges such as independence, active community participation, safety, housing choice, loneliness, social isolation, service accessibility, leisure, food, transportation, and agriculture until retirement. Conversely, seniors in rural settings may enjoy good health and quality of life, particularly in the cognitive [1] aspect, with better access to health services and healthier living and eating habits, although facing constant frailty risks [2].

Well-conducted cross-sectional and observational studies play a crucial role in aging research, measuring prevalence of health outcomes, understanding social determinants of health, and describing population characteristics [3].

There remains a need for a synthesis of aging in the rural world from the perspective of cognition, physical function, and life purpose. Thus, this systematic review protocol addresses the question: [1] What is the prevalence of cognition, physical functioning, and life purpose in older adults aging in rural community settings?

## Methods and analysis

This systematic review protocol adheres to the PICO strategy and is registered with the International Prospective Register of Systematic Reviews (PROSPERO) under number 2022 CRD42022311053.

### Eligibility criteria

**Databases and search strategy.** The search will include cross-sectional studies until April 2023 in journals indexed in health databases: Embase, MEDLINE, LILACS, PsycINFO, SciVerse Scopus (Scopus), SciELO, and Web of Science. Studies epidemiologically evaluating outcomes of cognition, functionality, and life purpose within rural aging and published in Portuguese, English, or Spanish until April 2023 will be accepted. The protocol follows Preferred Reporting Items for Systematic Reviews (PRISMA) guidelines [4].

**Search methods.** The search strategy will use 3-step Boolean operators and include terms such as "rural aging," "elderly," "old," "physical functioning," "cognition," and "life purpose." The entire search strategy for this systematic review was planned by a librarian and can be found in S1 Appendix. Two independent reviewers (HLMC and EBDL) will conduct the search, with a third reviewer included if tie-breaking criteria are needed.

**Study selection and data extraction.** After evaluating titles and abstracts from the searches, potential full texts will be assessed for eligibility by two independent reviewers. Authors of possible full texts will be contacted for questions about eligibility criteria. Studies meeting criteria will be included. For studies with the same sample, only those with the most representative example of that population will be considered.

Selected articles will extract data relevant to the theme, including year of publication, authors, characteristics of study institutions, sampling strategy, type of sample, prevalence of outcomes, and associated factors. Information will be organized and presented in tables.

**Study evaluation.** Methodological quality and risk of bias will be evaluated using the Observational Study Quality Evaluation (OSQE), widely used in the literature for cross-sectional studies [5]. Studies with scores higher than 5 will be included in the discussion.

## Strategy for data overview

The synthesis of outcome study data will adhere to the Meta-analysis of Observational Studies in Epidemiology (MOOSE) statement [5, 6]. When two or more articles report results from the outcome database, only the most comprehensive one will be included in the meta-analysis. Results will be presented in Forest Plots, showing 95% confidence intervals and p-values. Heterogeneity will be assessed using the $I^2$ statistic, considering it high when $I^2$ equals or exceeds 75%. Statistical procedures will be performed in STATA 14.0, with a significance level set at 5% for two-tailed tests [6].

There is no direct patient or public involvement in this study.

Ethical approval is not required, as no personal or private information of individuals will be involved. The intention is to submit this study to a peer-reviewed academic journal.

Cross-sectional studies, while economically feasible and easy to conduct, provide preliminary evidence to support more advanced studies. They offer insights into prevalence and incidence, generating essential hypotheses, and can establish possible associations with exposures or risk factors, whether analytical or descriptive [3].

The functional physical capacity of older people aging in rural settings appears better understood, with older women of good incomes more likely to be sedentary than older men [6]. Studies demonstrate factors associated with frailty in rural elderly, including age, gender, health status variables, self-perception of health, number of chronic conditions, disability in basic activities of daily living (ABVD), disability in instrumental ADLs, length of stay in the chair, and psychosocial problems. Depressive symptoms and cognitive impairment are notable, with increased comorbidity and disability in this population [7].

Regarding cognition, literature presents divergent views on the impact of the rural environment on aging. Sometimes, it is viewed positively [2], while at other times, not [8, 9].

The purpose of life in rural elderly individuals is still understudied, with limited knowledge about its impact on rural aging. Current literature suggests that the elderly with a purpose of life are more resilient, have less risk for developing dementia, feel less pain and are happier and more functional [10].

There are many questions to be answered about the way of aging in the rural context, and some gaps still need to be filled and understood to understand rural elder aging. After all, is rural aging positive or negative on the cognition, physical function, and life purpose of the elderly population?

## Supporting information

**S1 Checklist. PRISMA-P checklist.**
(PDF)

**S1 Appendix. Complete research strategy.**
(DOCX)

## Author Contributions

**Conceptualization:** Hércules Lázaro Morais Campos, Elisa Brosina De Leon, Ingred Merllin Batista de Souza, Elizabete Regina Araújo de Oliveira.

**Data curation:** Hércules Lázaro Morais Campos.

**Formal analysis:** Hércules Lázaro Morais Campos, Ingred Merllin Batista de Souza, Anna Quialheiro.

**Funding acquisition:** Hércules Lázaro Morais Campos.

**Investigation:** Hércules Lázaro Morais Campos, Elisa Brosina De Leon.

**Methodology:** Hércules Lázaro Morais Campos, Ingred Merllin Batista de Souza, Anna Quialheiro.

**Project administration:** Hércules Lázaro Morais Campos.

**Writing – original draft:** Hércules Lázaro Morais Campos, Elisa Brosina De Leon, Ingred Merllin Batista de Souza, Anna Quialheiro, Elizabete Regina Araújo de Oliveira.

**Writing – review & editing:** Hércules Lázaro Morais Campos, Elisa Brosina De Leon, Ingred Merllin Batista de Souza, Anna Quialheiro, Elizabete Regina Araújo de Oliveira.

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
