## [Decision Letter · Decision Letter 0]

1 Dec 2023

PONE-D-23-25450Cognition, physical function and life purpose in the rural elderly population: a systematic review protocolPLOS ONE

Dear Dr. Morais Campos,

Thank you for submitting your manuscript to PLOS ONE. After careful consideration, we feel that it has merit but does not fully meet PLOS ONE’s publication criteria as it currently stands. Therefore, we invite you to submit a revised version of the manuscript that addresses the points raised during the review process. Please submit your revised manuscript by the Jan 15 2024 11:59PM. If you will need more time than this to complete your revisions, please reply to this message or contact the journal office at plosone@plos.org. Please include the following items when submitting your revised manuscript:A rebuttal letter that responds to each point raised by the academic editor and reviewer(s). You should upload this letter as a separate file labeled 'Response to Reviewers'.A marked-up copy of your manuscript that highlights changes made to the original version. You should upload this as a separate file labeled 'Revised Manuscript with Track Changes'.An unmarked version of your revised paper without tracked changes. You should upload this as a separate file labeled 'Manuscript'.

We look forward to receiving your revised manuscript.

Kind regards,

Mahmoud M Werfalli, PhD

Academic Editor

PLOS ONE

When you resubmit, please ensure that you provide the correct grant numbers for the awards you received for your study in the ‘Funding Information’ section."

Reviewers' comments:

Reviewer's Responses to Questions

**Comments to the Author**

1. Does the manuscript provide a valid rationale for the proposed study, with clearly identified and justified research questions?

Reviewer #1: Partly

Reviewer #2: Yes

2. Is the protocol technically sound and planned in a manner that will lead to a meaningful outcome and allow testing the stated hypotheses?

Reviewer #1: Yes

Reviewer #2: Yes

3. Is the methodology feasible and described in sufficient detail to allow the work to be replicable?

Reviewer #1: No

Reviewer #2: Yes

4. Have the authors described where all data underlying the findings will be made available when the study is complete?

Reviewer #1: No

Reviewer #2: No

5. Is the manuscript presented in an intelligible fashion and written in standard English?

Reviewer #1: No

Reviewer #2: Yes

6. Review Comments to the Author

You may also provide optional suggestions and comments to authors that they might find helpful in planning their study.

Reviewer #1: It is challenging and complex to analyze 3 outcomes in only one systematic review. Elderly is not well accepted term; you may use “older” instead. Title is too generic; the outcome can be specified (prevalence?). The search limit (until April 2022 but incongruent with the one presented in the main text) should be updated. COSMIN aims to improve the selection of outcome measurement instruments both in research and in clinical practice by developing methodology and practical tools for selecting the most suitable outcome measurement instrument. Therefore it is not valid for the purpose of this study. A valid one could be: “Quality Assessment Tool for Observational Cohort and Cross Sectional Studies” from the National Institutes of Health. I find not necessary to specify the MESH terms in the Abstract. Perhaps this sentence then is not any more necessary (because repetitive): “Publication bias will be assessed using cross-sectional analytical study quality”. The Intro is too brief; the topic can be more deeply presented as well as the justification of the study (what is already known or published? Similar reviews?). You should consider stating in methods that you followed PRISMA-P. The search equations for every database should be presented. Regarding inclusion criteria, how “rural elderly” will be classified/identified?

The authors can describe where all data underlying the findings will be made available when the study is complete.

The manuscript needs a through English review.

Reviewer #2: I am grateful for the opportunity to review a study with such a relevant topic.

I suggest that the search strategy specified per database be added.

I suggest that the authors describe where all the data underlying the findings will be made available when the study is completed.

Yours sincerely.

7. PLOS authors have the option to publish the peer review history of their article (what does this mean?). If published, this will include your full peer review and any attached files.

Reviewer #1: No

Reviewer #2: **Yes: **Marcus Fernando da Silva Praxedes

---

## [Author Response · Author response to Decision Letter 0]

19 Feb 2024

Greetings

Here is our article with all the changes that have been suggested.

Yours sincerely.

Hércules Lázaro Morais Campos

---

## [Decision Letter · Decision Letter 1]

26 Apr 2024

Cognição, função física e propósito de vida na população idosa rural: protocolo de revisão sistemática

PONE-D-23-25450R1

Dear Dr. Prof. Hércules Campos

We’re pleased to inform you that your manuscript has been judged scientifically suitable for publication and will be formally accepted for publication once it meets all outstanding technical requirements.

Kind regards,

Mahmoud M Werfalli, PhD

Academic Editor

PLOS ONE

Reviewers' comments:

Reviewer's Responses to Questions

**Comments to the Author**

1. Does the manuscript provide a valid rationale for the proposed study, with clearly identified and justified research questions?

Reviewer #3: Yes

2. Is the protocol technically sound and planned in a manner that will lead to a meaningful outcome and allow testing the stated hypotheses?

Reviewer #3: Yes

3. Is the methodology feasible and described in sufficient detail to allow the work to be replicable?

Reviewer #3: Yes

4. Have the authors described where all data underlying the findings will be made available when the study is complete?

Reviewer #3: Yes

5. Is the manuscript presented in an intelligible fashion and written in standard English?

Reviewer #3: Yes

6. Review Comments to the Author

You may also provide optional suggestions and comments to authors that they might find helpful in planning their study.

Reviewer #3: The modifications made by the authors between the original version and the revised version are sufficient to ensure a high standard of rigor and transparency.

7. PLOS authors have the option to publish the peer review history of their article (what does this mean?). If published, this will include your full peer review and any attached files.

Reviewer #3: No

---

## [Editor Report · Acceptance letter]

16 May 2024

PONE-D-23-25450R1 

PLOS ONE

Dear Dr. Morais Campos, 

I'm pleased to inform you that your manuscript has been deemed suitable for publication in PLOS ONE. Congratulations! Your manuscript is now being handed over to our production team.

Kind regards, 

on behalf of

Dr. Mahmoud M Werfalli 

Academic Editor

PLOS ONE